# The Cancer-Associated Fibroblasts-Related Gene COMP Is a Novel Predictor for Prognosis and Immunotherapy Efficacy and Is Correlated with M2 Macrophage Infiltration in Colon Cancer

**DOI:** 10.3390/biom13010062

**Published:** 2022-12-28

**Authors:** He Ma, Qingqing Qiu, Dan Tan, Qiaofeng Chen, Yaping Liu, Bing Chen, Mingliang Wang

**Affiliations:** 1Department of General Surgery, RuiJin Hospital Lu Wan Branch, Shanghai Jiaotong University School of Medicine, Shanghai 200020, China; 2Central Laboratory, RuiJin Hospital Lu Wan Branch, Shanghai Jiaotong University School of Medicine, Shanghai 200020, China; 3Shanghai Institute of Hematology, State Key Laboratory of Medical Genomics, National Research Center for Translational Medicine at Shanghai, Ruijin Hospital, Shanghai Jiao Tong University School of Medicine, Shanghai 200025, China; 4Department of General Surgery, RuiJin Hospital, Shanghai Jiaotong University School of Medicine, Shanghai 200025, China

**Keywords:** colon cancer, COMP, cancer-associated fibroblasts, prognosis, immunotherapy

## Abstract

Background: Colon cancer is characterized by a sophisticated tumor microenvironment (TME). Cancer-associated fibroblasts (CAFs), which make up the majority of the stromal cells in TME, participate in tumor development and immune regulation. Further investigations of CAFs would facilitate an in-depth understanding of its role in colon cancer TME. Methods: In this study, we estimated CAF abundance based on The Cancer Genome Atlas (TCGA) and Gene Expression Omnibus (GEO) databases using the Microenvironment Cell Populations-counter (MCP-counter) algorithm. CAF-related genes were identified by differential gene expression analysis combined with weighted gene coexpression network analysis. For further selection, the least absolute shrinkage and selection operator (LASSO)-Cox regression was used, and the prognostic value of the selected gene was confirmed in numerous external cohorts. The function enrichment, immunological characteristics, tumor mutation signature, immunotherapy response, and drug sensitivity of the selected gene were subsequently explored. The bioinformatics analysis results were validated using immunohistochemistry on clinical samples from our institution. Results: According to our findings, cartilage oligomeric matrix protein (COMP) was uncovered as a candidate CAFs-driven biomarker in colon cancer and plays an important role in predicting prognosis in colon cancer. COMP upregulation was associated with enhanced stromal and immune activation, and immune cell infiltration, especially M2 macrophages. Genes that mutated differently between the high- and low-COMP expression subgroups may be correlated with TME change. Following verification, COMP reliably predicted the immunotherapy response and drug response. In addition, our experimental validation demonstrated that COMP overexpression is associated with colon cancer carcinogenesis and is strongly associated with CAFs and M2 macrophage infiltration. Conclusion: Our study uncovered that COMP was a key CAFs-driven gene associated with M2 macrophage infiltration and acted as a convincing predictor for prognosis and immunotherapy response in colon cancer patients.

## 1. Introduction

Colon cancer is the third most prevalent cancer and one of the top causes of cancer-related mortality globally [1]. Recently, colon cancer has presented increasing incidence and mortality among younger cases [2]. Despite the development of comprehensive and diverse therapeutic techniques, 30–50% of colon cancer patients have recurrence or metastases within five years of therapy [3]. Immunotherapy is an innovative treatment for colon cancer, but only a minority of patients exhibit microsatellite instability or increased tumor mutational burden (TMB), which are predictive biomarkers of immunotherapeutic response, which indicates that immunotherapies still fail to benefit many patients [4,5]. Therefore, searching for new immunotherapy targets appropriate for colon cancer patients is urgent.

Colon cancer is characterized by significant genetic and phenotypic heterogeneity, as well as a complex tumor microenvironment (TME) that includes admixtures of stromal cells, immune cells, vasculature, and other acellular extracellular matrix (ECM) components [6]. It was discovered that CAFs, which make up the majority of the stromal cellular components of the TME, induct epithelial–mesenchymal transition (EMT), maintain the stemness of cancer cells, and promote treatment resistance [7,8]. In addition, recent findings have identified CAFs as key immune regulators that modulate the tumor microenvironment of numerous malignancies, including colon cancer, through the release of several cytokines and chemokines, such as interleukin-1 (IL-1), interleukin-6 (IL-6), C-C motif chemokine ligand 2 (CCL2), and granulocyte-macrophage colony-stimulating factor (GM-CSF) [9,10]. A high abundance of CAFs in colon cancer could induce tumor inflammatory infiltration and promote angiogenesis, thus leading to a poor prognosis [11]. CAFs could also inhibit the activity of cytotoxic lymphocytes and induce the obliteration of CD8+ T cells, too [12,13]. Tumor-associated macrophages, also known as M2 macrophages, have been shown in TME to induce immunosuppression and tumor progression. CAFs have also been observed in colorectal cancer to enhance M2 macrophage infiltration and inhibit natural killer (NK) cell activity [14]. Given the importance of CAFs to the carcinogenic process and the tumor immune microenvironment, targeting strategies for CAFs may improve antitumor therapeutic efficacy. Considering that CAFs are highly heterogeneous, despite the fact that several candidate markers such as α-smooth muscle actin (α-SAM), also known as actin alpha 2 (ACTA2), fibroblast activation protein (FAP), and periostin (POSTN) have been identified and commonly used in colon cancer, it is still challenging to precisely define CAFs populations [15]. Among these markers, anti-CAF therapies have been mainly focused on eliminating FAP-positive CAFs to promote CD8+ T cells infiltration, thereby reigniting antitumor immunity and inhibiting tumor development [16]. Nevertheless, lethal bone toxicity and cachexia were observed in FAP+ stromal-cells-depleted murine models, and in clinical trials, anti-FAP inhibitors showed no therapeutic benefit in patients with metastatic colorectal cancer [17,18]. In addition, preclinical studies have reported the massive potential of targeting signaling pathways involved in CAFs activation or targeting CAFs-secreted factors, such as transforming growth factor-β (TGF-β), fibroblast growth factor (FGF), and IL-6, in cancer treatment, especially in immunotherapy. However, the therapeutic efficacy should be further evaluated in subsequent clinical trials [19,20]. Accordingly, research on the CAFs-related biomarkers may aid in elucidating the fundamental processes of carcinogenesis and finding more promising treatment options for colon cancer patients, individually.

In this study, we estimated CAFs’ abundance based on massive RNA-seq data from TCGA GEO datasets utilizing the MCP-counter algorithm, a typical technique for quantifying cell types in TME [21]. We sought to research the novel possible prognostic and immunotherapeutic biomarkers associated with CAFs and to investigate their biological roles in colon cancer. COMP was uncovered and validated as a candidate biomarker. As a secreted protein, COMP could promote cell proliferation, prevent apoptosis, and enhance EMT in colon cancer [22,23,24], whereas the significance of COMP in TME and immunotherapy has rarely been studied. Our current study elucidates the crucial role of COMP in infiltrating M2 macrophages, predicting the prognosis and immunotherapy response of individuals with colon cancer. Hence, COMP could act as a novel therapeutic target as well as a promising predictor for immune therapeutic response and prognosis in colon cancer.

## 2. Materials and Methods

### 2.1. Data Collection

TCGA RNA-seq data (normalized FPKM) and clinical profiles of colon adenocarcinoma (COAD) patients were harvested from the UCSC Xena database (https://xena.ucsc.edu, accessed on 20 June 2022) [25]. The information of somatic mutations (“maf” files) and copy number variation (CNV) (“SNP6” files) was obtained from Genomic Data Commons (GDC) data portal (https://portal.gdc.cancer.gov/, accessed on 7 June 2022). The gene profiles and clinical data of six colon and rectal adenocarcinoma cohorts, namely, GSE17536 [26], GSE39582 [27], GSE41258 [28], GSE17538 [26], GSE33113 [29], and GSE37892 [30], were collected from the GEO project https://www.ncbi.nlm.nih.gov/geo/, accessed on 11 June 2022). The GSE17538, GSE33113, and GSE37892 cohorts were integrated into a combined cohort, and the batch effects were adjusted with the ComBat algorithm in the R package “sva” (version 3.46.0) [31]. The training set consisted of 438 TCGA-COAD patients and 177 GSE17536 patients with complete follow-up information, while the GSE39582 (*n* = 556), GSE41258 (*n* = 182), and the combined GEO cohort (*n* = 406) were considered external validation datasets. The comparison of COMP expression in COAD and rectum adenocarcinoma (READ) versus normal samples were analyzed using the Gene Expression Profiling Interactive Analysis version 2 (GEPIA2) database (http://gepia2.cancer-pku.cn/, accessed on accessed on 7 July 2022) [32].

### 2.2. Calculation of CAFs Scores and Estimation of Immunological Features

The MCP-counter algorithm (https://github.com/ebecht/MCPcounter, accessed on 19 July 2022) [21] was performed to estimate the CAFs score. The immune, stromal, and ESTIMATE scores were assessed by the ESTIMATE algorithm (https://bioinformatics.mdanderson.org/estimate/, accessed on 20 August 2022) [33]. The tumor purity was inferred according to the ESTIMATE score. The infiltration of immune cells in TME was estimated by the CIBERSORT (https://cibersortx.stanford.edu/, accessed on 20 August 2022), quanTIseq (http://icbi.at/quantiseq/, accessed on 23 August 2022), and xCell (http://xCell.ucsf.edu/, accessed on 25 August 2022) algorithms [34,35,36]. The tracking tumor immunophenotype (TIP) approach provides rapid insights into the anticancer immune activity [37]. Xu et al. showed us the gene sets associated with the cancer immune cycle on a website (http://biocc.hrbmu.edu.cn/TIP/, accessed on 30 August 2022) [38]. Furthermore, the gene sets related to the response to immunotherapy were obtained from previous research [39]. To quantify the enrichment scores, the single sample gene set enrichment analysis (ssGSEA) derived from the gene set variation analysis (GSVA) R package (version 1.42.0, “ssgsea” function) was performed, and *p*-value < 0.05 was considered statistically different [40].

### 2.3. Differentially Expressed Gene (DEG) Analysis and Weighted Gene Coexpression Network Analysis (WGCNA)

We divided the samples from both the TCGA and GSE17536 cohorts into two CAFs subgroups on the basis of the medium CAFs score. DEG analysis between the two subgroups was applied utilizing the R package “limma” (version 4.4.4), and DEGs were considered as genes whose false discovery rate (FDR) value was <0.05 and |Log2 (fold change (FC))| > 1. WGCNA [41] was utilized to identify coexpressed gene modules with the most significantly related CAFs. The overlap of DEGs and CAFs-related gene modules revealed the hub genes.

### 2.4. Least Absolute Shrinkage and Selection Operator (LASSO)-Cox Regression and Candidate Selection

To identify the candidate gene, we further used LASSO-Cox regression, which is a regularization and descending dimension method for biomarker screening. Based on the hub genes, LASSO-Cox regression was applied, utilizing the “glmnet” R package (version 4.1-3) in the TCGA and GSE17536 cohorts, respectively. The intersection of genes selected at minimum values of lambda in the TCGA and GSE17536 cohorts was finally considered the candidate gene.

### 2.5. Functional Enrichment Analysis

DEGs between the high- and low-COMP subpopulations were identified using the R package “limma” (version 4.4.4). Gene Ontology (GO) analysis and Kyoto Encyclopedia of Genes and Genomes (KEGG) analysis of COMP-associated DEGs were applied with the “clusterProfiler” R package (version 3.18.0) to enrich relevant pathways, and then visualized in Metascape (https://metascape.org/, accessed on 2 September 2022) [42,43].

### 2.6. Prognostic Analysis and Nomogram Establishment

Analyses of overall survival (OS), disease-specific survival (DSS), disease-free survival (DFS), and progress-free survival (PFS) were conducted using Kaplan–Meier (K–M) curve (R package “survminer”, version 0.4.7) with log-rank tests. Univariate and multivariate Cox regressions were applied to explore the prognostic efficacy of COMP. A nomogram was developed using the “rms” R package on the basis of COMP and other independent prognostic indicators to predict the OS of colon cancer patients. Calibration curves were generated to assessed the predictive accuracy of the nomogram. Additionally, we used time-dependent receiver operating characteristic (ROC) curves (R package “timeROC”, version 0.4) to compare the prognostic value of this nomogram and tumor stage in different datasets. The area under the curve (AUC) was obtained by the R package “pROC” (version 1.18.0).

### 2.7. Analysis and Visualization of Somatic Mutations and CNV

The R package “maftools” (version 2.14.0) was applied to analyze the mutational patterns of TCGA samples [44]. Significantly different gene mutations between the low- and high-COMP subgroups and the mutual exclusivity and co-occurrence of gene mutations were analyzed by the “somaticInteractions” function. Twelve genes that significantly mutated in the high-COMP subgroup were inputted into the Gene Set Cancer Analysis (GSCA) [45] database (http://bioinfo.life.hust.edu.cn/GSCA/#/, accessed on 5 September 2022) to estimate the relation of immune cell infiltration and the gene set mutation level. TMB was calculated using the “tmb” function in “maftools”. The microsatellite instability (MSI) statuses of TCGA samples were harvested from the UCSC Xena database (https://xena.ucsc.edu, accessed on 20 June 2022). Moreover, Genomic Identification of Significant Targets In Cancer version 2.0 (GISTIC2, https://genepattern.broadinstitute.org/, accessed on 7 September 2022) [46] was utilized to analyze amplification and deletion and calculate G-scores with the input of “SNP6” files.

### 2.8. Immunotherapeutic Response Prediction and Drug Sensitivity Exploration

The TIDE algorithm is a machine learning approach that could predict the immune-checkpoint blockade (ICB) response [47]. Herein, we leveraged TIDE analysis (http://tide.dfci.harvard.edu/, accessed on 9 September 2022) to evaluate potential responses to ICB in the TCGA, GSE17536, GSE39582, and GSE41258 cohorts. Using the Biomarker Exploration for Solid Tumors (BEST) database (https://rookieutopia.com/, accessed on 13 September 2022), we explored three external cohorts (Wolf cohort [48], Van cohort [49], and IMvigor210 cohort [39]) to verify the effect of COMP expression on the responses of the patients receiving anti-PD-L1 treatment. We also retrieved clinical and RNA-seq data from a molecular analysis based on samples in the phase 3 JAVELIN Renal 101 trial (NCT02684006) (https://clinicaltrials.gov/ct2/show/NCT02684006, accessed on 13 September 2022) of avelumab plus axitinib versus sunitinib in advanced renal cell carcinoma (aRCC) [50] to compare the different prognoses in high- and low-COMP subgroups of patients in the trial.

To investigate the potential agents for colon cancer patients with different COMP expression levels, the sensitivity scores were yielded using the R package “oncoPredict” (version 0.2) [51] to predict the half-maximal inhibitory concentration (IC50) of all drugs in the Genomics of Drug Sensitivity in Cancer v2 (GDSC2) database (https://www.cancerrxgene.org/, accessed on 17 September 2022) [52]. The Connectivity Map (CMap) database (https://clue.io/, accessed on 20 September 2022) was a comprehensive resource to search for potential agents and their underlying mechanisms [53]. We queried CMap to screen potential agents by comparing the similarity of DEGs between high- and low-COMP subgroups with database signatures. The query result showed compounds ranked with enrichment scores and we selected the top 50 relative compounds that might be capable of targeting COMP.

### 2.9. Sample Collection and Immunohistochemical (IHC) Staining Evaluation

As a validation, 40 colon carcinoma specimens with 27 corresponding adjacent normal tissues were collected from patients diagnosed at Ruijin Hospital Lu Wan Branch, Shanghai Jiao Tong University School of Medicine. The patients underwent surgical resection between 2019 and 2021, and all tumors were primary and untreated before surgery. For each patient, the clinicopathologic characteristics, including age, gender, tumor location, tumor size, tumor differentiation, tumor stage, and MSI status, were available. Tumors were assessed according to the AJCC staging system for colorectal cancer (8th). Among the 40 patients, tumor and blood samples from 34 patients were analyzed by the ABI3730 analyzer (Applied Biosystems Inc.) to detect MSI status.

COMP, CD206, and ACTA2 expression levels were detected using IHC. Paraffin-embedded tissues were cut into slices of 5 μm thickness. Tissue sections were incubated with anti-COMP antibody (1:100, ab231977, Abcam, Cambridge, UK), anti-CD206 antibody (1:3000, ab252921, Abcam, Cambridge, UK), or anti-ACTA2 antibody (1:30,000, ab7817, Abcam, Cambridge, UK). The images were captured using a panoramic scanner PANNORA-MIC (3DHISTECH, Budapest, Hungary) with software CaseViewer 2.4 (3DHISTECH). The staining intensity and rate of positive cells were analyzed using the software AIpathwell (Servicebio, Wuhan, China), and histochemistry score (H-Score = ∑ (PI × I) = (percentage of cells of weak intensity × 1) + (percentage of cells of moderate intensity × 2) + percentage of cells of strong intensity × 3) [54] was utilized to semi-quantify the protein expression levels. H-score < 100, H-score of 100–200, and H-score > 200 were considered weak, low-, and high-positive staining, respectively.

### 2.10. Statistical Analysis

Statistical analyses were conducted using R software v4.1.3 (Auckland, New Zealand) and its appropriate packages. The optimal cutoff point for continuous variables was determined utilizing the R package “survminer” in this study. Comparison between groups was conducted utilizing the Student’s *t*-test for normally distributed variables and the Mann–Whitney U-test for nonnormally distributed variables. The Wilcoxon matched-pairs signed rank test was used to compare matched samples for nonparametric data. Depending on the data normality, the Pearson’s or Spearman’s correlation analysis was used to determine the linear relationship between variables. Fisher’s exact test was used to analyze correlations between the COMP expression subgroups and clinicopathologic characteristics. Two-sided *p*-value < 0.05 was considered statistically significant.

## 3. Results

### 3.1. The Abundance of CAFs Was Correlated with the Progression and Immune Landscape of Colon Cancer

A detailed flowchart of this study is shown in Figure 1A. Applying MCP-counter, the abundance of CAFs was estimated in the TCGA and GSE17536 datasets. In the TCGA cohort, CAFs were more prevalent than other cell types in the TME (Figure 1B) and exhibited a strong connection with the stromal score, immune score, ESTIMATE score, and tumor purity (Appendix A). Similar results were obtained in the GSE17536 dataset (Appendix A). Furthermore, the abundance of CAFs is primarily enhanced in colon cancer patients with a more advanced N stage (*p* = 0.010) in the TCGA cohort (Figure 1B). The high-CAFs subtype was linked to a poor prognosis of PFS survival (*p* = 0.002) in the TCGA cohort, but the prognosis of OS survival (*p* = 0.144) did not differ between the two CAFs subtypes (Figure 1C). In the GSE17536 cohorts, the high-CAFs subtype correlated with a poor prognosis for both OS (*p* < 0.001) and DFS (*p* = 0.001) survival (Figure 1D). The above results suggest the correlation of an enhanced abundance of CAFs with high immune, stromal, and ESTIMATE score, low tumor purity, and a poor prognosis in colon cancer patients.

### 3.2. COMP Was Identified as a Key Gene Related to CAFs in Colon Cancer Patients

Between the two CAFs subgroups, the volcano plot showed 323 DEGs and 71 DEGs in the TCGA and GSE17536 cohorts, respectively (Figure 2A). In the WGCNA, 5 and 4 were selected as the optimal soft thresholds in the TCGA and GSE17536 cohort, respectively (Appendix A). According to WGCNA results, the turquoise modules in both the TCGA and GSE17536 cohorts were recognized as the most significant modules associated with CAFs (Figure 2B). At the confluence of DEGs and turquoise gene modules, 45 hub genes were discovered (Figure 2C) and subsequently inputted into LASSO-Cox regression. Finally, COMP were outputted from the TCGA cohort, whereas COMP, THBS2, COL1A2, GREM1, SPOCK1, HTRA1, NEXN, OLR1, ADAM12, SERPINF1, and CTSK were outputted from the GSE17436 cohort (Figure 2D,E). Herein, we selected COMP, which was the intersection of the output results from the two training cohorts, as the key gene associated with CAFs in colon cancer patients.

We further performed GSEA between high- and low-COMP subgroups in the TCGA dataset. GO enrichment analysis revealed that COMP was involved in extracellular matrix organization, cell–substrate adhesion, cell–cell adhesion, response to growth factor, and regulation of MAPK cascade (Appendix A). KEGG analysis enriched pathways linked to focal adhesion, regulation of actin cytoskeleton, calcium signaling pathway, TGF beta signaling pathway, ECM receptor interaction, and MTOR signaling pathway (Appendix A).

### 3.3. Prognostic Value of COMP in Colon Cancer Patients

Firstly, considering that tumors originating in the colon and rectal tissues shared strong molecular similarities, we compared COMP expression in COAD and READ versus normal tissues, utilizing the GEPIA2 dataset, and found that both COAD and READ had a higher COMP expression than normal tissues (Figure 3A). Furthermore, patients with higher T, N, and TNM stage had higher COMP expression in the TCGA cohort (Figure 3B). Moreover, the prognostic significance of COMP was validated in the TCGA, GSE17536, GSE39582, GSE41258, and the combined-GEO cohort, showing that high-COMP expression was associated with worse prognosis (Appendix A). Meanwhile, through univariate and multivariate Cox regression, COMP’s prognostic value was evaluated. A higher COMP expression signified a poor prognosis in the five datasets. After adjusting for clinicopathologic characteristics, the COMP expression was discovered as an independent risk factor in the five datasets except for DFS in the GSE39582 dataset (Figure 3C).

To better predict the prognosis of colon cancer patients, a nomogram was constructed by integrating age, gender, TNM stage, and COMP expression, and the nomogram score was estimated to forecast the 1-, 3-, and 5-year OS for individuals with colon cancer (Figure 3D). The calibration plot demonstrated that the nomogram accurately predicted patient OS based on the ideal model (Figure 3E). Meanwhile, the time-dependent AUC indicated a better performance of the nomogram-based signature in OS prediction than the tumor stage in the TCGA, GSE17536, and GSE39582 cohorts (Figure 3F).

### 3.4. COMP Was Related to the Immune Signature in Colon Cancer

We calculated CAFs abundance, immune scores, stromal scores, and tumor purity utilizing the MCP-counter and ESTIMATE algorithms and compared them between the COMP subgroups. Immune and stromal scores were notably higher in the high-COMP subgroup, while tumor purity was lower (Appendix A). The CIBERSORT algorithm was further employed to assess the fractions of 22 immune cells in each patient from the TCGA and GSE17536 cohorts, and we illustrated the differentially infiltrating patterns across different COMP subgroups (Appendix A). The different infiltration of immune cells between the two COMP expression subgroups was also investigated. In the TCGA dataset, our results showed that the fractions of M0 and M2 macrophages were considerably higher in the high-COMP subgroup, whereas M1 macrophages as well as CD8+ T cells showed no significant distinction between the two subgroups (Appendix A). In the GSE17536 cohort, M0 and M2 macrophages were also shown to have higher fractions in the high-COMP subgroup, whereas a significantly lower level of CD8+ T cells was recognized (Appendix A). Meanwhile, correlation analyses showed a positive correlation of COMP expression with the M2 macrophage fraction, and a negative correlation with CD8+ T cell fraction (Figure 4A). The infiltrations of immune cells were also assessed by the xCell and quanTIseq algorithm, and the positive correlations of M2 macrophages with COMP expression were observed in both the TCGA and GSE17536 cohorts (Figure 4B).

Furthermore, we investigated the expression of macrophage polarization-related chemokines and cytokines in the low- and high-COMP subtypes [55,56]. It was observed that M2 macrophage chemokines (IL-10, CSF1, TGF-β1, TGF-β2, and TGF-β3) were extensively expressed in the high-COMP subtype, whereas M1 macrophage chemokines (TNF, HMGB1, IFNG, and CSF2) showed no significant difference between the two subtypes (Figure 4C). According to these findings, patients in the high-COMP subgroup were more inclined to display an M2 phenotype. Subsequently, the connections of COMP expression with the cancer immunity cycle and signaling pathways associated with ICB treatment were investigated. Our findings indicated that COMP correlated positively with the majority of cancer immunity cycle phases, while inversely with almost all ICB-treatment-related signaling pathways (Figure 4D).

### 3.5. Association of COMP with Mutational Landscape in Colon Cancer

As revealed from the mutation analysis, the occurrence of somatic mutations was evaluated in the low- and high-COMP subsets in the TCGA cohort, and the top 15 frequently mutated genes were exhibited (Appendix A). Nevertheless, no prominent differences in somatic mutations of these genes were investigated in the low- and high-COMP expression categories. Furthermore, maftools analysis results showed the top 15 genes that mutated differently between the two COMP subgroups (Figure 5A). Additionally, mutation co-occurrences were observed in most of the 15 differently mutated genes (Figure 5B). Among these genes, twelve were mutated more in the high-COMP group, including ROS1, SPTBN1, DSCAM, LRP2, DNAH7, SVEP1, GOLGB1, KIF21A, LTBP2, PKD1L1, DNAH8, and SYNE2. Interestingly, we found that patients with the mutations of ROS1, SVEP1, and DNAH7 had a poor prognosis in the TCGA cohort (Figure 5C). Additionally, we inputted the twelve genes as a gene set into the GSCA database and found that the mutation subgroup had lower proportion of naïve CD8+ cells and higher abundance of macrophages, whereas the proportion of CD8+ T cells showed no significant difference between the mutation and WT subgroups (Appendix A). Subsequently, we calculated the TMB and curated MSI status and identified a negative correlation between COMP and TMB and a reduced COMP expression in the MSI-H group (Appendix A). We further calculated G-score to compare somatic CNV levels between the two COMP expression subgroups and found that the low-COMP subgroup displayed higher levels of CNV, and the mutations occurred in more regions, than the high-COMP expression subgroups (Figure 5D,E).

### 3.6. COMP Expression Could Predict the Clinical Benefit of ICB

To further clarify the role of COMP in immunotherapy, we explored the associations between COMP and the TIDE score, which is an accurate predictor for ICB therapies [47]. A significant positive correlation of COMP expression with the TIDE score was found in the TCGA datasets. According to the TIDE algorithm, the low-COMP expression group showed considerable therapeutic benefits and more response to immunotherapy. Meanwhile, COMP expression was considerably enhanced in ICB nonresponders (Figure 6A). There was substantial concordance in the GSE17536, GSE39582, and GSE41258 cohorts (Appendix A). On the basis of the BEST database, we further investigated the association between COMP expression and the benefits of immunotherapy in three external cohorts receiving anti-PD-L1 treatment (Wolf cohort, Van cohort, and IMvigor210 cohort). In the Wolf cohort, a higher expression of COMP was also detected in nonresponders, and the AUC value was 0.651, indicating that COMP is capable of distinguishing responders from nonresponders who received anti-PD-L1 treatment (Figure 6B). However, significant differences in COMP expression were not observed between the responders and nonresponders in the Van cohort (Figure 6C) and the IMvigor210 cohort (Appendix A). In the JAVELIN Renal 101 trial, patients receiving avelumab plus axitinib showed a PFS benefit versus sunitinib in patients with aRCC (Appendix A). Interestingly, based on the transcriptome and prognosis data, we did not find the benefit in the high-COMP expression subgroup (Figure 6D). However, in the low-COMP expression subgroup, avelumab plus axitinib exhibited a better PFS than sunitinib in aRCC patients (Figure 6E). All these results revealed that patients with low expression of COMP could benefit more from ICB treatment.

### 3.7. Evaluation of Drug Sensitivity and Identification of Potential Therapeutic Drugs Targeting COMP

To find potential therapeutic compounds, the oncoPredict package was applied to assess the sensitivity of agents in the GDSC2 database. The IC50 value was quantified using the sensitivity score, and the top 10 sensitive chemotherapeutic drugs for low- and high-COMP subgroups were illustrated (Figure 7A). The patients in the high-COMP subgroup exhibited a higher sensitivity to BMS-754807, JQ1, SB216763, Staurosporine, Dasatinib, AZ960, AZD1332, WZ4003, IGF1R_3810, and JAK_8517, whereas the patients in the low-COMP subgroup displayed a higher sensitivity to Vinblastine, Cytarabine, Oxaliplatin, Gemcitabine, YK-4-279, Eg5_9814, AZD5991, KRAS (G12C) Inhibitor-12, Uprosertib, and vinorelbine.

The CMap database featured a vast gene expression profile that could examine gene expression, phenotype, and pharmacological linkages and offer insight on mechanisms of action. We displayed the top 50 compounds targeting COMP and depicted their potential mechanisms from the mode-of-action (MoA) analysis. The result revealed that five compounds (panobinostat, SB-939, HC-toxin, BRD-K67506692, and apicidin) shared the MoA of HDAC inhibitor. Meanwhile, five compounds shared the MoA of protein synthesis inhibitor and four compounds shared the MoA of retinoid receptor agonist, followed by VEGFR inhibitor with three compounds (Figure 7B).

### 3.8. Validation of the COMP Signature with Colon Cancer Samples from Our Institution

We further performed IHC staining of COMP, CAFs marker ACTA2, and M2 macrophage marker CD206 for 40 surgical specimens in our institution. Our findings revealed that the COMP expression levels were elevated in advanced T, N, and AJCC stage, but not related to age, gender, tumor location, tumor size, differentiation, N stage, and MSI status. Meanwhile, a high COMP positive rate was identified in a higher T, N, M, and AJCC stage (Table 1). Moreover, the coexpression of COMP, ACTA2, and CD206 was found in colon cancer specimens, especially in the high-COMP expression subgroup (Figure 8A,B). The correlation analysis revealed the COMP expression level to be positively associated with ACTA2 and CD206, respectively (Figure 8C,D). We also detected COMP expression levels between colon carcinoma and adjacent normal tissues and found an elevated COMP expression level in carcinoma specimens (Figure 8E,F).

## 4. Discussion

Immunotherapy provides a novel approach for the treatment of colon cancer, but only a subset of patients benefit from the ICB targeting therapy [57]. Recent studies reveal that CAFs provide resistance to immunotherapy via interaction with immune cells and immune components in the TME [9]. In the present study, we explored the critical genes associated with CAFs abundance through analysis of the TCGA and GEO cohorts. Our results showed the potential of COMP as an independent prognosis factor and biomarker for predicting immunotherapeutic response and efficacy for colon cancer patients.

Previous research demonstrated that CAFs are abundant in colon cancer patients with a bad prognosis [11]. Consistently, we also detected that CAFs correlated with a worse outcome as well as a high immune and stromal score, indicating that CAFs interacted with both stromal and immunological elements. Utilizing LASSO-Cox regression to screen the intersection of the DEGs and WGCNA gene modules associated with CAFs, we uncovered COMP as the candidate gene in both the TCGA and GSE17536 cohorts. COMP is a secretory protein consisting of five subunits of homologous glycoprotein and has a variety of biological roles in multiple cell types and tissues [58]. COMP has been reported to be implicated in the EMT and other well-known cancer cell signaling pathways, such intracellular calcium homeostasis, Notch3, Akt, MEK/ERK, etc. [22,23,24,59,60]. Interestingly, we found that COMP exhibited essential involvements in response to growth factor, regulation of MAPK cascade, calcium signaling pathway, and ECM receptor interaction. COMP has been shown to be highly expressed in tumor cells and the neighboring stroma cells in breast, hepatocellular, and colon cancer [22,61,62]. We also detected that COMP expression was increased in colon cancer and exhibited a stage-dependent increase, and the similar expression profile was also validated by IHC in specimens from our institution. Previous studies reported the survival predicting value of COMP in colorectal cancer [24], while, in this study, systematic analysis of various independent colon cancer cohorts revealed that high COMP expression indicated a worse outcome. The stratified examination indicated that the COMP was able to anticipate survival outcomes independently of other clinical characteristics. Additionally, this study developed the nomogram, and the outstanding precision of the nomogram was then validated by time-dependent AUC in different cohorts. Taken together, our study evaluated COMP’s robust performance in predicting colon cancer prognosis and suggests that COMP might have practical usages as a complement for the tumor stage in colon cancer.

Emerging studies reveal that distinct TME characteristics may be associated with distinct prognoses and varying degrees of immunotherapy efficacy [6,7]. Our findings showed that COMP was highly associated with CAF abundance and stromal score, which was followed by the validation of COMP coexpression with ACTA2 in clinical samples, indicating that COMP expression may partially reflect CAF abundance in colon cancer. It is considered that tumor-associated macrophages can polarize into two major phenotypes: proinflammatory M1 and protumorigenic M2 [63]. Our findings revealed that high-COMP subgroup patients were prone to exhibit the M2 phenotype, and IHC results also showed a coexpression of COMP and the M2 macrophages marker. Studies also show that CSF-1, TGF-β, and IL-10, secreted by CAFs, are required for monocytes to differentiate into M2-like TAMs [55,56]. Consistently, several chemokines and cytokines that induced the M2 phenotype were shown in our result. These findings imply that the COMP- and other CAFs-released molecules, collectively, promoted an M2 immunosuppression phenotype in colon cancer TME, which might contribute to a poor prognosis. Although the immunomodulatory processes in TME are intricate, the cancer immunity cycle is regarded as a thorough representation of the overall effect of these processes [37]. Herein, we found that COMP was favorably connected with the activities of numerous phases in the cancer immune cycle and inversely connected with immunotherapy-related signs. All these results indicate that COMP-induced M2 macrophage infiltration impaired the high activity of antitumor immune processes, and the treatment targeting COMP may reverse the M2 immunosuppression phenotype and be instrumental for immunotherapy in colon cancer patients.

The progression and efficacy of immunotherapy for colon cancer is also closely associated with TMB, MSI status, and chromosomal instability [5]. We found that among the 15 genes most mutated in the high-COMP subgroup, ROS1, SVEP1, and DNAH7 mutations result in a worse prognosis in colon cancer patients. More recently, the rearrangements of ROS1 have been recognized as a therapeutic predictor for colorectal patients in the MSS subgroup [64]. Interestingly, the mutation of these genes also enhanced the infiltration of macrophages and impaired the infiltration of CD8+ naïve cells, and these genes exhibited a high incidence of co-mutations, suggesting that the co-occurrence of mutations in these genes may result in an unidentified modification in TME. The influence of these co-mutations on TME and immunotherapy efficacy warrants further investigation. Further analysis revealed that COMP was negatively correlated with TMB, and COMP expression was lower in patients with MSI-H. Moreover, higher frequencies of amplification and deletion were noted in the low-COMP expression subpopulation; considering the lower stromal and immune scores, the low-COMP group displayed an immunity-deprived phenotype, which may receive more benefits from immunotherapy and cisplatin treatment [65]. Indeed, based on the TIDE score and several external anti-PD-L1 treatment cohorts, patients in low-COMP expression subgroups indicated a better response to immunotherapy. We next sought to explore potential therapeutic compounds targeting COMP. Considering the toxicity of directly targeting CAFs markers, such as FAP, recent studies have paid more attention to targeting the CAFs-secreted factors or activation pathways. For instance, targeting CAFs-derived CXCL12 with the administration of AMD3100, the inhibitor of the CXCL12 receptor, revealed the antitumor effects in pancreatic cancer [66]. Furthermore, active breast CAFs could secrete high levels of IL-6, whereas the IL-6 receptor inhibitor tocilizumab, via inhibiting the IL-6/STAT3/AUF1 pathway, could reverse active breast CAFs and reduce their paracrine procarcinogenic activities [67]. Additionally, TGF-β and FGF are key determinants of fibroblast activation and proliferation. Targeting the TGF-β pathway with galunisertib or targeting the FGF receptor with erdafitinib could inhibit CAFs activation and promote antitumor immunity [68,69]. In this study, the drug sensitivity assessment showed that some drugs, such as staurosporine, dasatinib, vinblastine, cytarabine, and oxaliplatin, had different responses in the two COMP expression subgroups. We also analyzed the CMap database and proposed HDAC inhibitor, protein synthesis inhibitor, retinoid receptor agonist, and VEGFR inhibitor for targeting COMP in colon cancer. These findings indicate that taking COMP expression into consideration might be helpful in therapeutic applications. Overall, COMP could be a valuable biomarker for prognosis prediction and administering treatments, including chemotherapy, targeted therapy, and immunotherapy.

Despite the fact that an integrative analysis based on independent datasets was undertaken and certain findings with potential values for therapeutic application were achieved, the role of COMP in the prognosis and immunotherapy for colon cancer should be prospectively validated. In addition, the role of COMP in predicting immunotherapy responses for colon cancer patients was assessed using the TIDE algorithm and verified in several ICB treatment cohorts, such as breast cancer and aRCC; however, we did not conduct the analysis in colon cancer patients receiving immunotherapy because of the absence of transcriptome data. Lastly, although the coexpression of COMP with ACTA2 and CD206 was found in this study, it should be validated in a more extensive cohort to avoid the selection bias, and additional validation through in vitro and in vivo studies needs to be addressed.

In summary, our study uncovered COMP as a critical CAFs-driven gene in colon cancer, which was externally and extendedly validated in multiple independent cohorts. Our findings indicated that COMP was related to M2 macrophage infiltration and acted as a convincing predictor of prognosis and immunotherapy in colon cancer patients.

## Figures and Tables

**Figure 1 biomolecules-13-00062-f001:**
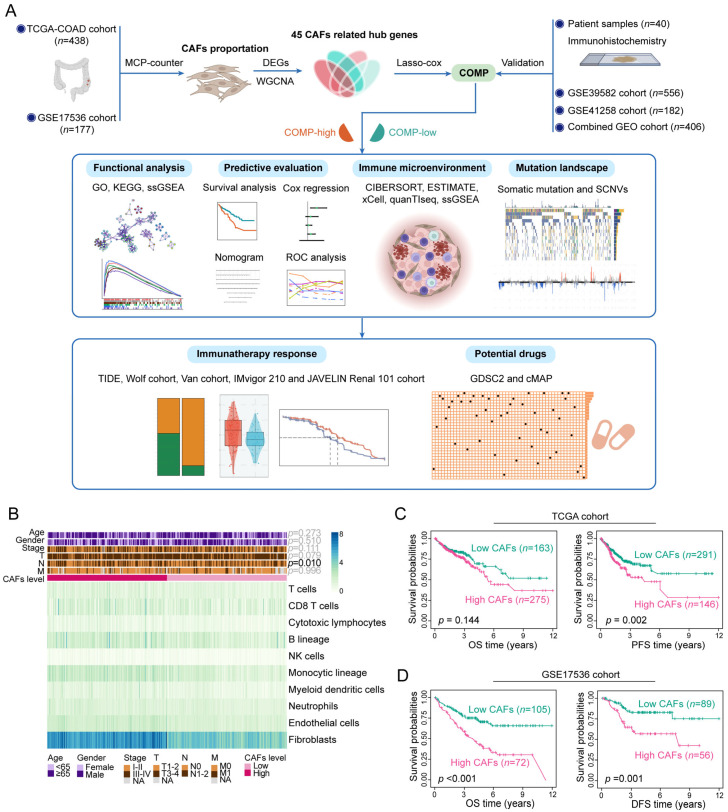
Assessment of clinical features and the immune landscape associated with CAFs. (**A**) The whole flowchart of the study design; (**B**) Various TME cell abundances in the TCGA cohort are shown in the heat map. Associations between CAFs level and clinicopathological characteristics are also illustrated as an annotation; (**C**) K–M plots of OS and PFS for two CAFs subtypes in the TCGA dataset; (**D**) K–M plots of OS and DFS for two CAFs subtypes in the GSE17536 dataset.

**Figure 2 biomolecules-13-00062-f002:**
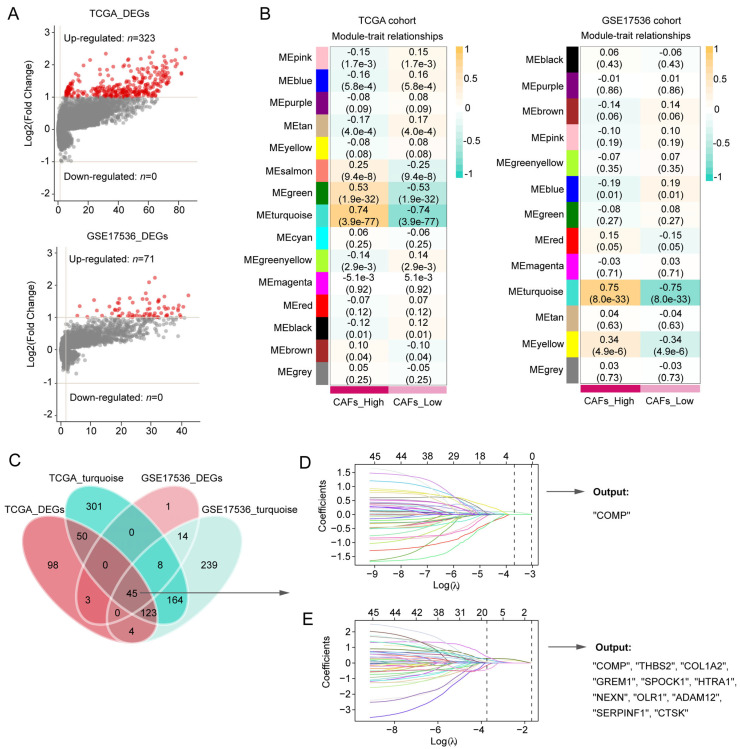
Screening for CAFs-associated genes in the TCGA and GSE17536 datasets and identification of COMP as a key gene. (**A**,**B**) CAFs-associated DEG analysis and gene modules identified by WGCNA in the TCGA and GSE17536 datasets, respectively. The turquoise module in both TCGA and GSE17536 displayed the closest relationship with CAFs; (**C**) Venn plot shows 45 hub genes intersected by DEG and WGCNA analyses. The hub genes were subsequently screened using the LASSO-Cox regression model in the (**D**) TCGA and (**E**) GSE17536 datasets, respectively. The intersection of the output genes revealed COMP as a key CAFs-related gene.

**Figure 3 biomolecules-13-00062-f003:**
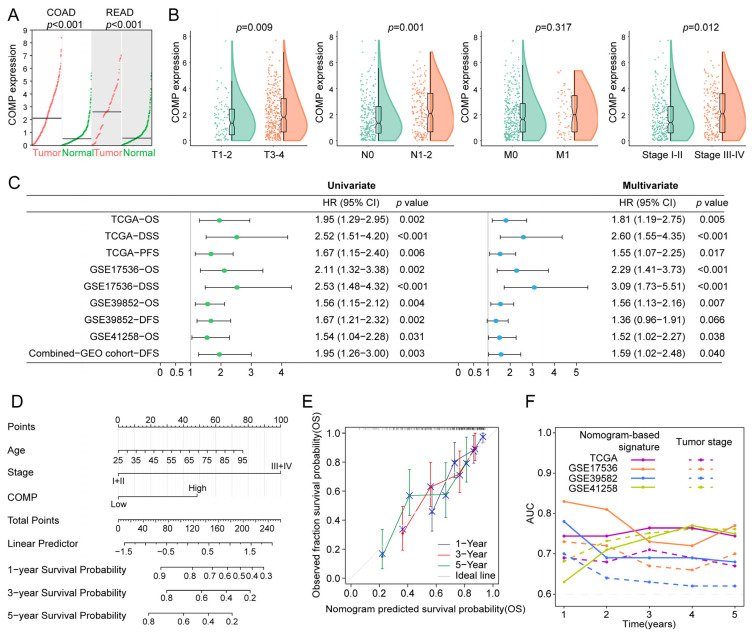
Evaluation and validation of the prognostic value of COMP. (**A**) Analysis of COMP expression between tumor and normal control specimens in COAD and READ; (**B**) Comparison of COMP expression in different T, N, M, and TNM stages; (**C**) Forest plot of Cox analysis. For multivariate analysis in the TCGA dataset, the HR value of COMP level was adjusted by age, gender, and T stage. As for GSE17536, GSE39582, GSE41258, and the combined-GEO dataset, HR value of COMP level was adjusted by age, gender, TNM stage, and grade; age, gender, and TNM stage; age and gender; age, gender, and TNM stage, respectively; (**D**) Nomogram for predicting the OS of patients with colon cancer; (**E**) Calibration curves display the agreement between predicted OS and actual survival duration; (**F**) Time-dependent AUC plotted for different durations of OS for nomogram-based signature and tumor stage in the TCGA, GSE17536, GSE39582, and GSE41258 datasets.

**Figure 4 biomolecules-13-00062-f004:**
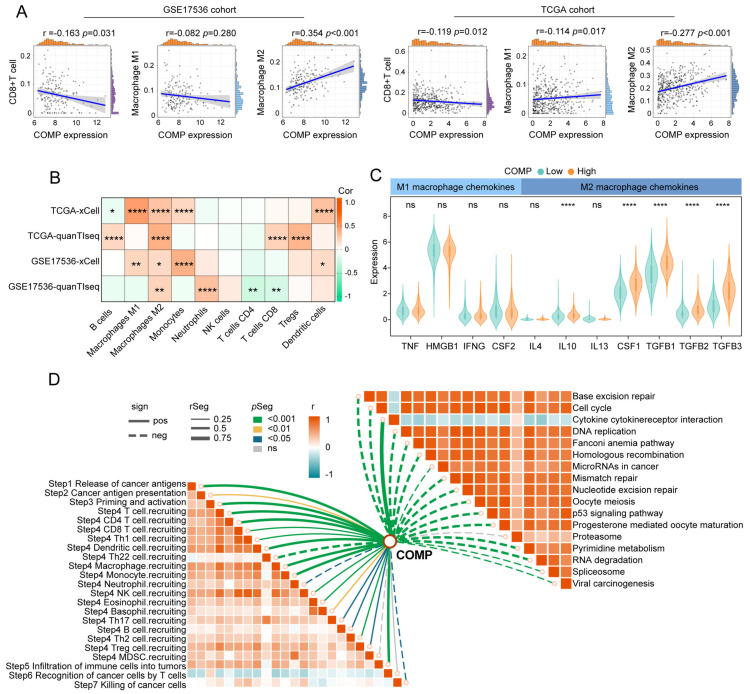
The immune landscape of COMP in colon cancer. (**A**) Correlations between COMP expression with CD8+T cells, M1 and M2 macrophage infiltration levels in the TCGA and GSE17536 datasets; (**B**) Heat map illustrating correlations between COMP expression and the infiltration levels of 10 immune cells estimated using xCell and quanTIseq algorithms; (**C**) Expression of representative macrophage chemokines between low- and high-COMP expression subgroups; (**D**) Associations of COMP expression with the enrichment scores of immunotherapy-related pathways and the steps of the cancer immunity cycle (* *p* < 0.05; ** *p* < 0.01; **** *p* < 0.0001; ns, not significant).

**Figure 5 biomolecules-13-00062-f005:**
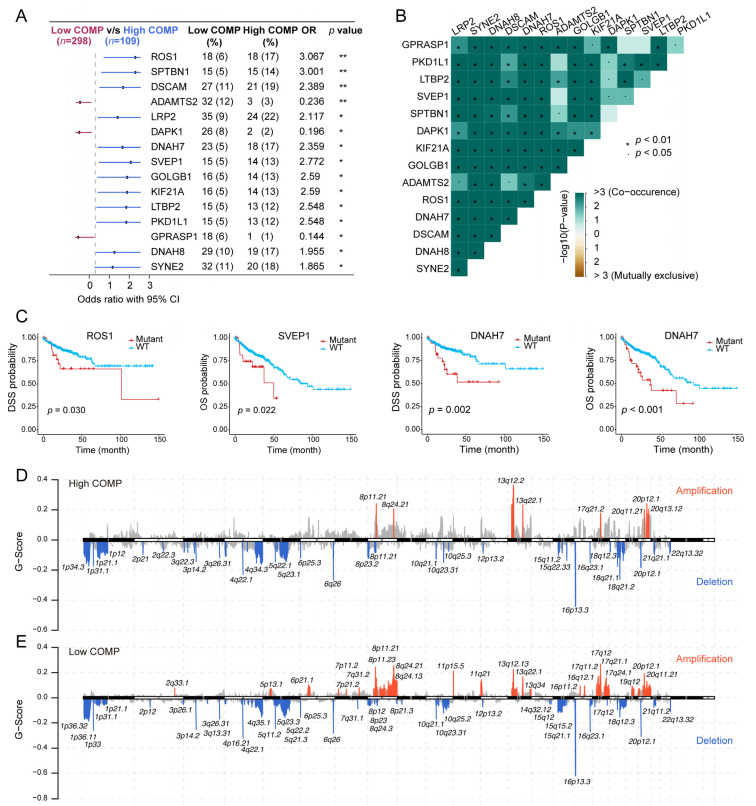
Correlation of COMP with the mutation landscape. (**A**) Forest plot of genes mutating differentially between low- and high-COMP expression subgroups; (**B**) Mutual exclusivity and co-occurrence analysis of genes mutating differentially between the two subgroups; (**C**) K–M curves display that patients with ROS1, SVEP1, and DNAH7 mutations had a poor prognosis; (**D**,**E**) Comparison of amplification and deletion of copy numbers in high- and low-COMP expression subgroups.

**Figure 6 biomolecules-13-00062-f006:**
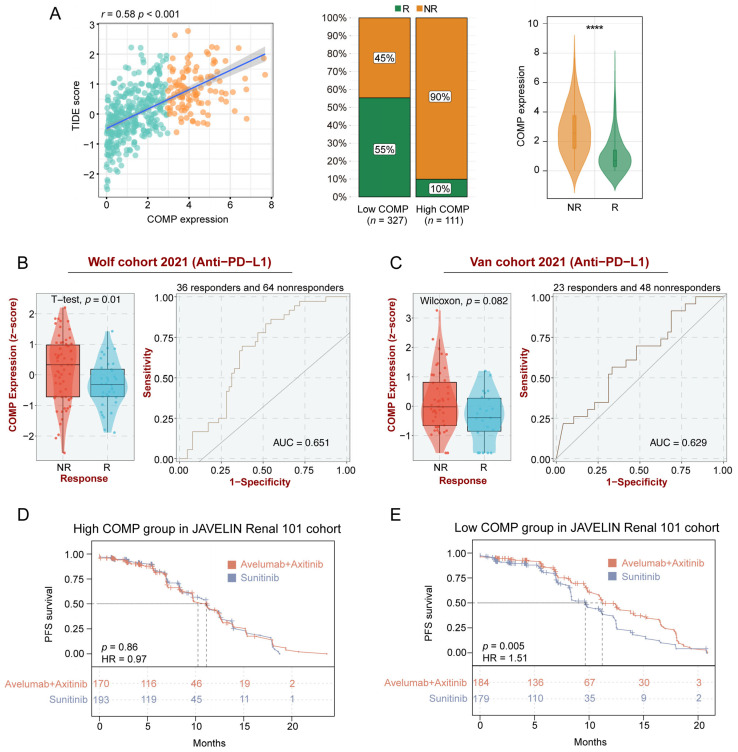
Identification of COMP expression for predicting immunotherapy responses. (**A**) The correlations between COMP expression and the TIDE score, the proportions of patients with putative immunotherapy response in two COMP expression subgroups, and the comparisons of COMP expression between distinct responses in the TCGA cohort. R, responder; NR, nonresponder; (**B**,**C**) Evaluation of COMP expression in anti-PD-L1 responders versus nonresponders and ROC curves of COMP expression in the Wolf and Van cohorts; (**D**,**E**) K–M plots for aRCC patients treated with the combination of avelumab and axitinib versus sunitinib alone in high- and low-COMP expression subgroups from the JAVELIN Renal 101 cohort (**** *p* < 0.0001).

**Figure 7 biomolecules-13-00062-f007:**
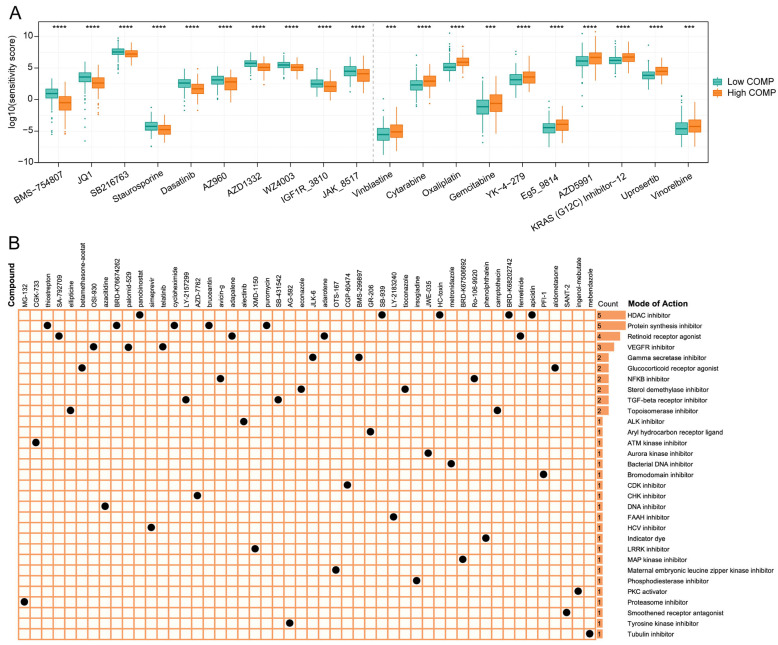
Association between COMP expression and drug sensitivity. (**A**) Predicted sensitivity scores of candidate potent drug for high- and low-COMP expression subgroups (BMS-754807, fold change = 0.437; JQ1, fold change = 0.622; SB216763, fold change = 0.806; Staurosporine, fold change = 0.757; Dasatinib, fold change = 0.624; AZ960, fold change = 0.738; AZD1332, fold change = 0.684; WZ4003, fold change = 0.782; IGF1R_3810, fold change = 0.730; JAK_8517, fold change = 0.767; Vinblastine, fold change = 1.611; Cytarabine, fold change = 1.584, Oxaliplatin, fold change = 1.510; Gemcitabine, fold change = 1.741; YK-4-279, fold change = 1.566; Eg5_9814, fold change = 1.545; AZD5991, fold change = 1.560; KRAS (G12C) Inhibitor-12, fold change = 1.449; Uprosertib, fold change = 1.471; Vinorelbine, fold change = 1.448); (**B**) The heat map displays compounds and their shared MoA based on the CMap database (*** *p* < 0.001; **** *p* < 0.0001).

**Figure 8 biomolecules-13-00062-f008:**
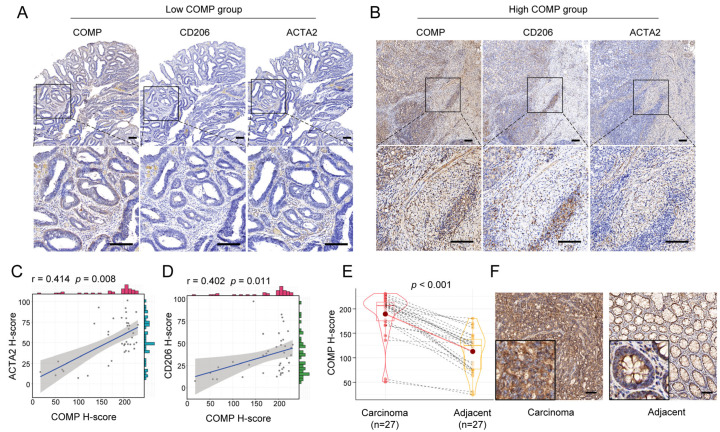
Verification of the correlation between COMP expression and infiltration levels of CAFs and M2 macrophages in colon cancer specimens. (**A**,**B**) Coexpression analysis of COMP with the CAFs marker ACTA2 and the M2 macrophage marker CD206 using IHC staining in low- and high-COMP expression subgroups; (**C**) Correlation between the expressions of COMP and ACTA2; (**D**) Correlation between the expression of COMP and CD206. (**E**,**F**) Comparison of COMP expression in primary colon cancer and adjacent normal tissues by IHC staining (scale bar = 100 μm).

**Table 1 biomolecules-13-00062-t001:** Correlation between COMP expression and pathological parameters in colon cancer.

Characteristics.	Cases	H-Score Median	*p*-Value ^1^	Categorized H-Score		High Positive Rate (%)	*p*-Value ^2^
				Weak or Low	High		
**Age**							
<60	15 (37.5%)	204.67		5	10	25.00	
≥60	25 (62.5%)	202.34	0.586	11	14	35.00	0.740
**Gender**							
Female	18 (45%)	199.08		9	9	22.50	
Male	22 (55%)	206.37	0.097	7	15	37.50	0.335
**Tumor location**							
Left-sided	26 (65%)	204.43		11	15	37.50	
Right-sided	14 (35%)	203.84	0.630	5	9	22.50	0.746
**Tumor size**							
<5 cm	25 (62.5%)	202.34		11	14	35.00	
≥5 cm	15 (37.5%)	205.57	0.150	5	10	25.00	0.740
**Differentiation**							
Poor	12 (30%)	189.39		6	6	15.00	
Well to moderate	28 (70%)	204.43	0.535	10	18	45.00	0.490
**Stage**							
I–II	19 (47.5%)	173.45		13	6	15.00	
III–IV	21 (52.5%)	208.05	0.010 *	3	18	45.00	<0.001 *
**T**							
1–2	7 (17.5%)	135.65		6	1	2.50	
3–4	33 (82.5%)	205.57	0.025 *	10	23	57.50	0.011 *
**N**							
0	18 (45%)	169.95		13	5	12.50	
1–2	22 (55%)	207.42	0.006 *	3	19	47.50	<0.001 *
**M**							
0	25 (62.5%)	195.5		14	11	27.50	
1	15 (37.5%)	208.33	0.086	2	13	32.50	0.010 *
**MSI status**							
MSI-H	3 (8.82%)	210.98		0	3	7.50	
MSS/MSI-L	31 (91.18%)	204.67	0.172	11	20	50.00	0.535

^1^*p*-values were calculated using Mann–Whitney U-test. ^2^*p*-values were calculated using Fisher’s exact test (* *p* < 0.05).

## Data Availability

The names and accession numbers of the public datasets presented in this study can be found in Materials and Methods.

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
