# Peer review of "The Cancer-Associated Fibroblasts-Related Gene COMP Is a Novel Predictor for Prognosis and Immunotherapy Efficacy and Is Correlated with M2 Macrophage Infiltration in Colon Cancer"

_biomolecules, 2022, doi:10.3390/biom13010062_

Round 1
Reviewer 1 Report
Colon cancer is a type of cancer that begins in the large intestine (colon). Immunotherapy provides a solution for the treatment of colon cancer, but only a subset of patients benefits from the ICB targeting therapy. CAFs provide resistance to immunotherapy via interaction with immune cells and immune components in the TME.
The authors have detected that CAFs correlated with a worse outcome as well as a high immune and stromal score, indicating that CAFs interacted with both stromal and immunological elements.
In this research work, the authors have estimated CAFs’ abundance based on The Cancer Genome Atlas (TCGA) and Gene Expression Omnibus (GEO) databases using the MCP-counter algorithm. LASSO-Cox regression was calculated and the prognostic value of the selected gene was confirmed in numerous external cohorts.
They have explored the function enrichment, immunological characteristics, tumor mutation signature, immunotherapy response and drug sensitivity of the selected gene.
They found that, cartilage oligomeric matrix protein (COMP) was uncovered as a candidate CAFs-driven biomarker in colon cancer. COMP upregulation was associated with enhanced stromal and immune activation, immune cell infiltration, especially M2 macrophages. Genes mutated differently between the high- and low-COMP expression subgroups may be correlated with TME change. Following verification, COMP reliably predicted the immunotherapy response and drug response.
The experimental results support their claim and conclusion. So, I recommend the paper for publication in its present form.
Author Response
Dear Reviewer,
Thanks very much for taking your time to review this manuscript. We really appreciate you for the positive comments.
Our main conclusion has not changed, but we hope the new text makes the significance of these data much clearer.
Reviewer 2 Report
Colon cancer is a dangerous tumor type, and relevant biomarkers might improve the therapy and prognosis. Cancer-associated fibroblasts make up a significant portion of the tumor microenvironment and could be one of the targets in cancer therapy. The manuscript focuses on the description of a new biomarker associated with CAFs in colon cancer, and the translational potential of this secreted protein, COMP. There are several points mentioned below that could be considered to further improve the manuscript presentation.
Specific points.
1. Lines 67-68. The sentence seems to be incomplete, kindly check.
2. Figure 1, and panel A, in particular, seems to be very complex. One option could be to simplify it, to warranty that all the labels are readable, or remove the labels and elements that are not possible to read. If the idea is to get a scheme, a detailed description of each element is not needed. If the idea is to provide specific information for each case, then the font should be larger, and readable, and the Figure could be divided, e.g., into several Figures and Supplementary elements.
3. Line 248. The template text is not removed. Kindly fix it here and through the text. "Figure 1. This is a figure. Schemes follow the same formatting. FIGURE 1"
4. Figure 2. The fonts are not readable. Kindly consider simplifying or dividing this Figure into several main and/or supplementary elements.
5. Figure 3 is too busy. The fonts are too small and hardly readable, as presented. The only way to read it is by increasing the size on a screen. The printed version will not be easy to follow.
6. Figure 4. Too busy and hardly readable due to the selected small font.
7. Line 367. "Significantly" seems to be a wrong word choice. It could be "significant" instead.
8. Line 367 refers to Figure 8. Usually, the Figures are mentioned one by one, meaning that Figures 6 and 7 must be referred to earlier than Figure 8 after Figure 5.
9. Figure 5. Too small font, it is hardly readable and busy Figure.
10. Figure 6. Busy and hardly readable Figure with small font. Consider simplifying or dividing. The font should be readable on the printed version of the manuscript.
11. Figure 7. It is about the right size, except that the font is relatively small and the presentation would benefit from a larger font.
12. Figure 7A. Log10 sensitivity score makes a visually small difference when the real difference is large, i.e. several folds. It could be beneficial to label or comment in Figure legend the statistical differences, if any, between COMP-low and COMP-high groups.
13. Line 471. Kindly fix units (micro), "(Scale bar = 100 um)".
14. The manuscript concludes that COMP is a key factor for the prognosis of immunotherapy efficiency in colon cancer patients. It would be beneficial for readers to place COMP in the context of what is known now regarding CAF-related biomarkers. The attempts to identify key factors to target CAFs are made for several years, and mentioning other factors and mechanisms both in the Introduction and Discussion would complement the current study and place COMP in the right perspective.
Author Response
Dear Reviewer,
We feel great thanks for your professional review work on our article. According to your nice suggestions, we have made extensive modifications to our manuscript and supplemented extra data. We hope our revisions will meet with your approval.
Please see my itemized responses in the attachment.
Thanks again!

Reviewer 3 Report
The authors provided a well-written bioinformatic and statistical paper but they also investigated and tested the results obtained in a small cohort.
The methods and the results are detailed and clear.
I have just a few points to be fixed:
-In some figures, the image seems not well compressed and some words result not completely clear. Please check that this is not an error in the figure creation.
-Line 248: "This is a figure. Schemes follow the same formatting. FIGURE 1" -> I think that this should be removed.
-Line 277: "pathway (Figure 2G)" -> Add a period at the end of the sentence.
-The legend of figure 3 continues under figure 4. This could be very misleading. Please check with the editor if it is possible to correct this.
-Figure 5C: The figure legend show a scale of colors from "Mutually exclusive" to "Co-occurrence" but none seems to be "Mutually exclusive". Is it correct?
-Line 279: The READ group is introduced without any explanation. Why these samples were added to the analysis? Are there any similarities with the COAD? Please add a few lines to explain better.
-Line 505: "colon cancer.." -> Please, remove a period.
-Line 570: "Supplementary Materials:" -> None of the 3 supplementary figures are cited in the main text. This should be fixed.
Author Response
Dear Reviewer,
Thank you for your comments and suggestion concerning our manuscript. The comments and suggestions are all valuable and very helpful for revising and improving our paper. We have studied comments carefully and have made correction which we hope meet with your approval.
Please see my point-by-point responses in the attachment.
Thanks again!
